# Polyphenolic and Fruit Colorimetric Analysis of Hungarian Sour Cherry Genebank Accessions

**Francesco Desiderio** [1], **Samuel Szilagyi** [1], **Zsuzsanna Békefi** [1], **Gábor Boronkay** [2], **Valentina Usenik** [3], **Biserka Milić** [4], **Cristina Mihali** [5] and **Liviu Giurgiulescu** [5,*]

1   Research Centre of Fruit Growing, Hungarian University of Agriculture and Life Sciences, 2100 Godollo, Hungary
2   Ornamental Plant and Green System Management Research Group, Hungarian University of Agriculture and Life Sciences, 2100 Godollo, Hungary
3   Department of Fruit Growing, Viticulture and Horticulture, University of Ljubljana, 1000 Ljubljana, Slovenia
4   Faculty of Agriculture, University of Novi Sad, 21102 Novi Sad, Serbia
5   Faculty of Science, Department of Chemistry-Biology, Technical University of Cluj–Napoca, 400114 Cluj–Napoca, Romania
*   Correspondence: giurgiulescu@gmail.com; Tel.: +40-74-031-0674

**Abstract:** Sour cherry is one of the most important horticultural crops in the Hungarian market. Its flavour combination makes it ideal for fresh consumption as well as canned products. The Hungarian and European markets have requested for new varieties to be introduced, making the evaluation of breeding and prebreeding material a crucial point. A total of 30 sour cherry accessions from the genebank collection were investigated for their potential inclusion into the breeding program. The main aim of the study was to identify candidates for future breeding programs, selecting their colour profiling and total polyphenolic content (TPC). This study follows the antioxidant activity of cherry species by determining the total content in polyphenols. Polyphenols are found in higher concentration in cherries when compared to other plants and have been identified as free radical scavengers, which are useful to prevent the occurrence of several diseases. Furthermore, TPC has been observed as a contributor of bitterness, acidity, colour, flavour, odour, and oxidative stability. The accessions were evaluated for their colour, fruit weight, flavour profile, firmness, and TPC. Colorimetric data were compared among four methods to illustrate to the Hungarian breeders which of the available approaches is the most accurate for sour cherry breeding. Results suggested that several accessions appear relevant for the breeding program, such as 'Pipacs 1', 'Bosnyák', 'Hortenzia Királynője', and 'Mogyoródi Kései'. The total polyphenolic content was highest in 'Pipacs1' and lowest in 'Kántorjánosi 3'. 'Pipacs 1' and 'Hortenzia Királynője' had interestingly high acidity content and light to very light fruit colours. 'Bosnyák' had a deep and dark colour with high soluble sugar content. 'Mogyoródi Kései' appears to have the biggest fruit. All mentioned varieties will be included in future breeding programs.

**Keywords:** *Prunus cerasus* L.; CIELab; genetic viability; total polyphenolic content

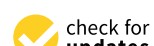



## 1. Introduction

Sour cherry (*Prunus cerasus* L.) is a valuable horticultural crop in Europe [1,2]. Hungary is one of the top 10 producers of sour cherry in Europe, with yields greater than 60 thousand tons/year [3]. The origin of sour cherry is thought to be around the area of the Caspian Sea and Black Sea [4], and is widely considered beneficial for human consumption. Its health benefits are numerous: cherry consumption can decrease hypertension and chronic inflammatory diseases, and it has been investigated for its anticarcinogenic benefits [5,6]. In a recent review, Kelley and colleagues mentioned that its low caloric content and high quantity of nutrients and bioactive food components, such as fibre, phenols, carotenoids, potassium and vitamin C, make it a very healthy fruit [5]. When compared

with the health benefits of other small fruits, sour cherry appears to have high levels of polyphenols [6]. Polyphenols are a widely known group of bioactive compounds, including flavonoids, phenolic acids, and anthocyanins. Polyphenolics are well known for their action against free radicals and their prevention of chronic diseases such as cancer, obesity, and cardiovascular diseases [7]. Hence, it appears that sour cherry is a natural source of health benefits. Hungarian varieties are well known globally, cultivated with different names, such as 'Danube' and 'Balaton', which are called 'Érdi Bőtermő' and 'Újfehértói fürtös' in Hungary [8]. The main characteristics of Hungarian sour cherries are the flavour balance, ratio of sweetness and acidity, and their colour. Sour cherries are used for both fresh and processed consumption such as juices, liqueurs and jams [9,10]. Darker colours are more appreciated in Europe and there is still demand for new varieties, particularly for those that can satisfy the fresh market with fruit that is larger and firm and that has attractive colour and balanced flavour [2,10–12]. For this reason, 30 varieties from the germplasm collection located in the University of Agriculture and Life Sciences (MATE) Research Institute of Fruit Growing were analysed for their colour and chemical properties. Due to their phenotype differences and variance, we hypothesize that the colour of sour cherry fruit might be correlated with different polyphenolic content. The main aim of our research was to evaluate the accessions and varieties according to their colour, physical characteristics, and total polyphenolic content to determine whether there are significant correlations between them, to then be able to select valuable material from the germplasm collection and include them in the breeding program. Furthermore, different methods for colourimetry were compared to help Hungarian sour cherry breeders better differentiate sour cherry fruit.

## 2. Materials and Methods

A total of 30 sour cherry accessions (Table 1) were selected for analysis from the genebank collection of the University of Agriculture and Life Sciences, Fruit Growing Research Station in Érd, near Budapest, Hungary (47°20′55.1″ N 18°51′48.1″ E). Fruits were collected at the ripe stage on the last days of June, depending on the accession and variety, in 2022. Of the 30 accessions, nine are commercially available cultivars from the selection of the Hungarian collection. Approximately 500 g of fruit were collected from four clonal trees of each accession, from the northern (N), southern (S), eastern (E), and western (W) directions starting at the upper and moving towards the lower branches. For each accession, 20 fruits were selected and used for the following analyses as replicates. Pictures were taken both at the research station and in the open field before collection (Supplementary Materials).

**Table 1.** Varieties selected from the germplasm collection located in the Fruit Growing Research station. Origin, parental lineage, and harvesting time are indicated for each accession. Asterisks (*) indicate the commercial cultivars.

| Accession | Origin | Parents | Harvest Time |
|---|---|---|---|
| Bagi Meggy | Carpathian Basin | Landrace | 29 June 2022 |
| Bosnyák | Carpathian Basin | Landrace | 29 June 2022 |
| Cigány Késői | Carpathian Basin | Landrace | 29 June 2022 |
| Cigánymeggy 7 * | Carpathian Basin | Landrace | 30 June 2022 |
| Dunabogdányi | Carpathian Basin | Landrace | 29 June 2022 |
| Édes Pipacs | Carpathian Basin | Landrace | 28 June 2022 |
| Érdi Bőtermő * | Hungary | Pándy × Nagy angol | 29 June 2022 |
| Érdi Jubileum * | Hungary | Pándy × Eugenia | 30 June 2022 |
| Fehérvári | Carpathian Basin | Landrace | 28 June 2022 |
| Fűzlevelű Kisszemű | Carpathian Basin | Landrace | 28 June 2022 |
| Helyi Sötét | Carpathian Basin | Landrace | 28 June 2022 |
| Hortenzia Királynője | France | Landrace (also called Königin Hortense) | 29 June 2022 |
| Kántorjánosi 3 * | Hungary | Landrace | 30 June 2022 |

**Table 1.** *Cont.*

| Accession | Origin | Parents | Harvest Time |
|---|---|---|---|
| Késői cigány | Carpathian Basin | Landrace | 28 June 2022 |
| Késői parasztmeggy | Carpathian Basin | Landrace | 28 June 2022 |
| Későn virágzó | Carpathian Basin | Landrace | 29 June 2022 |
| Korai cigány | Carpathian Basin | Landrace | 28 June 2022 |
| Korai Pándy | Carpathian Basin | Landrace | 29 June 2022 |
| Májusi hólyag | Carpathian Basin | Landrace | 29 June 2022 |
| Mogyoródi kései | Carpathian Basin | Landrace | 28 June 2022 |
| Nagy Gobet | France | Landrace (also called Grosse Gobet) | 29 June 2022 |
| Pándy 279 * | Carpathian Basin | Landrace | 30 June 2022 |
| Pándy 43 * | Carpathian Basin | Landrace | 29 June 2022 |
| Pándy Bb. 119 * | Carpathian Basin | Landrace | 30 June 2022 |
| Péceli nagy | Carpathian Basin | Landrace | 29 June 2022 |
| Pipacs1 * | Carpathian Basin | Landrace | 30 June 2022 |
| Szamosi meggy | Carpathian Basin | Landrace | 29 June 2022 |
| Tiszabög 50/7 | Carpathian Basin | Landrace | 29 June 2022 |
| Újfehértói fürtös * | Újfehértó, Hungary | Landrace | 30 June 2022 |
| Velencei kései | Carpathian Basin | Landrace | 28 June 2022 |

### 2.1. Firmness, Soluble Solid Content (SSC), Acidity, Sweetness, Juiciness, Fruit, and Seed Weight

Samples were analysed for firmness using a nondestructive durometer (Hardness tester flat tip, T.R. Turoni s.r.l., Forlì, Italy) and for soluble solid content (SSC) with a refractometer (HI-96801, Hannah Instrument Ltd., Leighton Buzzard, UK). Following the Union for the Protection of New Varieties of Plants (UPOV), guidelines for sour cherry (CPVO-TP/230/1) descriptors for acidity, sweetness, and juiciness were considered. Fruit acidity was rated from very low (1) to very high (9), sweetness from low (3) to high (7), and juiciness from weak (3) to strong (7). Fruit and seed weights were measured using a standard scale with an accuracy in the range of 0.01 g. Fruit pulp was calculated as the difference between fruit and seed weight.

### 2.2. Colour Scaling: Pantone, UPOV, CTIFL and CIELab

Fruit colour was measured in four comparable ways. The first was with CTIFL scale system (Centre Techique Interprofessionnel des Légumes, Rungis, France), commonly used by breeders. The scale varies from light pink (1) to black (7). The second was the Union for the Protection of New Varieties of Plants (UPOV) guidelines for sour cherries (CPVO-TP/230/1) for skin colour from orange red (1) to blackish (6). Pantone palette (Formula Guide Printer Edition, Coated Paper Printer colour management kit, Product Code: PANT013, Manufacturers #: GP1201, Ashford, UK) was used to record colours and converted according to the CIELab scaling system. Ultimately, colour was measured with a spectrophotometer CM-600d (Konica Minolta Inc., Osaka, Japan) and the CIELab method was followed, using D65 illuminant and a $10°$ observer angle. L* measures the lightness from 0 (black) to 100 (white), a* measures the red (+a*) to green ($-$a*) spectrum, and b* measures the yellow (+b*) to blue ($-$b*) intensity. Chroma ($C* = a^2 + b^2$) indicates the saturation or intensity of colour [13] and a modified hue angle ($h \pm 33°$) indicates the amount of redness to yellowness [$H° = \arctan (b*/a*)$], where $0°$ to $360°$ define colours from red to magenta, $90°$ to $180°$ from yellow to green, and $180°$ to $270°$ defines blue. The modified hue angle ($h^{\pm33°}$) was calculated with Colour Conversion Centre 4.1 (http://ccc.orgfree.com/, accessed on 15 May 2023). A total of ten fruits were measured for each accession, with 50 repeated measurements for each fruit, covering the surface in different measurement points. Measured samples were then kept at $-20$ °C for further chemical analysis.

### 2.3. Total Phenolic Content (TPC) Analysis

Total polyphenolic content (TPC) of cherries was analysed by the Folin–Ciocalteu colorimetric method according to Singleton and Rossi [14] with some slight changes. The samples (5 g) were mixed with 96% acidulate ethanol solution (30 mL) in a test tube. An ultrasonic water bath (Evo Sonic Ultrasonic Bath-POKA) was used for the extraction of phenolics for 30 min. After centrifugation of the extracts for 20 min, liquid samples were filtered through a 0.45 μm filter. The extracts (1 mL), Folin–Ciocalteu reagent (5 mL), and distilled water (0.7 mL) were mixed in a test tube. After 5 min incubation at 25 °C, 5 mL of sodium carbonate solution (7.5%) was added. Mixtures were incubated in a dark place for 2 h at 25 °C. The absorbances were measured at 750 nm by a UV-VIS spectrophotometer (Perkin-Elmer Lambda 850+, Milan, Italy). The analyses were carried out in triplicate and the results were given as the average value of these experiments. Gallic acid (GA) was used to draw the calibration curve and results were expressed for 1 g of dried sample as the mg GA equivalent (GAE).

### 2.4. Statistical Analysis

Data were tested for normal distribution. Afterwards, the analysis of variance (ANOVA) was performed in SPSS (SPSS software version 25, IBM®, Armonk, NY, USA), where L*, a*, b*, C*, and $h^{\pm 33°}$ were each analysed against total polyphenolic content (TPC). TPC was then compared to fruit firmness and soluble solid content (SSC) as well as fruit weight, seed weight, and pulp weight. Different colorimetric tests were contrasted according to the CIELab standard method. Pantone colorimetric scale, Ctifl, and UPOV skin colour were singularly compared to L*, a*, b*, C*, and $h^{\pm 33°}$. Post hoc tests were performed by using the Tukey's b test and considered significant at $p < 0.05$. Principal component analysis (PCA) was carried out between the colorimetric, TPC and fruit firmness, SSC, fruit, and pulp weight. Bivariate analysis was also performed, where Spearman's rho was considered significant at $p < 0.05$.

## 3. Results

### 3.1. Sour Cherry Colour Comparison

Colorimetric data were collected using different methods to compare which of those might be useful to Hungarian breeders. When comparing CIELab values with UPOV Skin colour, Pantone, and Ctifl, we observed significant differences between b*, Chroma, and hue for Ctifl ($p < 0.05$). The UPOV scaling system was significantly different for L*, b*, and hue ($p < 0.05$). Pantone colour was significantly different for all of L*, a*, b, Chroma, and hue ($p < 0.05$). UPOV skin colour scale system was in line with the CIELab scaling system, where the lowest value recorded was two and the highest was seven (Table 2). When compared, the Ctifl scaling system indicated that the lowest values of L*, a*, b*, and Chroma were in line with the CIELab chromatic scaling system (Table 3). The Pantone skin colour scale recorded the highest value as Pantone 1807 C and the lowest value as Pantone Black 6 C (Table 4).

**Table 2.** Skin colour scale for sour cherry (UPOV) compared with CIELab single values. Skin colour values range from orange red (1) to blackish (6). Letters of the same column indicate significantly different values at $p < 0.05$. Bold and italic values represent highest and lowest, respectively.

| Skin Colour (UPOV Scale) | L* | a* | b* | Chroma | $h^{\pm 33°}$ |
|---|---|---|---|---|---|
| 6 | **25.16** ± 0.90 a | **4.26** ± 2.54 a | **0.52** ± 0.88 a | **4.32** ± 2.65 a | **−29.29** ± 6.89 a |
| 5 | 26.29 ± 0.56 b | 9.21 ± 1.55 b | 2.10 ± 0.57 b | 9.45 ± 1.63 b | −20.33 ± 1.40 b |
| 4 | 26.67 ± 0.98 b | 13.04 ± 3.52 c | 3.43 ± 1.36 c | 13.49 ± 3.74 c | −18.71 ± 1.92 b,c |
| 3 | 27.11 ± 2.20 b | 15.59 ± 5.51 c | 4.72 ± 2.92 d | 16.32 ± 6.14 c | −17.24 ± 3.08 c |
| 2 | 29.02 ± 1.12 c | 18.93 ± 2.65 d | 6.92 ± 1.42 e | 20.16 ± 2.97 d | *−13.09* ± 1.35 d |
| 1 | *29.92* ± 0.94 c | *24.88* ± 2.41 e | *8.55* ± 1.33 f | *26.31* ± 2.71 e | −14.13 ± 1.17 d |

**Table 3.** Ctifl scale compared with CIELab values. Ctifl scale system range here from light red (2) to black (7). Letters of the same column indicate significantly different values at $p < 0.05$. Bold and italic values represent highest and lowest, respectively.

| Ctifl Scale | L* | a* | b* | Chroma | h$^{\pm33°}$ |
|---|---|---|---|---|---|
| 7 | *25.14* ± 0.93 a | *4.30* ± 2.26 a | *0.49* ± 0.72 a | *4.35* ± 2.32 a | *−29.23* ± 7.04 a |
| 6 | 25.86 ± 1.16 a,b | 6.70 ± 3.88 a | 1.38 ± 1.36 a | 6.87 ± 4.05 a | −25.01 ± 6.97 b |
| 5 | 26.63 ± 1.05 b | 12.37 ± 3.89 b | 3.19 ± 1.48 b | 12.79 ± 4.14 b | −19.07 ± 2.00 c |
| 4 | 26.79 ± 1.36 b | 14.25 ± 4.19 b | 3.99 ± 1.92 b | 14.81 ± 4.56 b | −17.98 ± 2.19 c |
| 3 | 28.11 ± 2.85 c | 18.50 ± 5.96 c | 6.30 ± 3.47 c | 19.59 ± 6.76 c | −15.35 ± 3.69 d |
| 2 | **29.83** ± 0.97 d | **23.35** ± 3.53 d | **8.16** ± 1.57 d | **24.75** ± 3.82 d | **−13.81** ± 1.37 d |

**Table 4.** Pantone scale compared with CIELab. Pantone colours indicated here represent a single colour combination. Letters of the same column indicate significantly different values at $p < 0.05$. Bold and italic values represent highest and lowest, respectively.

| Pantone Scale | L* | a* | b* | Chroma | h$^{\pm33°}$ |
|---|---|---|---|---|---|
| Pantone Black 6 C | *25.09* ± 1.02 a | *3.48* ± 1.77 a | *0.32* ± 0.64 a | *3.52* ± 1.82 a | *−30.58* ± 8.00 a |
| Pantone 222 C | 26.27 ± 0.56 b | 9.19 ± 1.44 b | 2.07 ± 0.51 b | 9.43 ± 1.51 b | −20.43 ± 1.33 b |
| Pantone 209 C | 26.49 ± 0.72 b | 13.35 ± 3.93 c,d | 3.56 ± 1.51 c,d | 13.82 ± 4.18 c,d | −18.55 ± 1.07 c |
| Pantone 202 C | 26.43 ± 0.86 b | 12.71 ± 2.21 c,d | 3.30 ± 0.83 c,d | 13.13 ± 2.34 c,d | −18.60 ± 2.05 c |
| Pantone 195 C | 26.41 ± 1.13 b | 12.14 ± 1.85 c | 3.14 ± 0.62 c | 12.54 ± 1.94 c | −18.63 ± 1.41 c |
| Pantone 193 C | 30.72 ± 1.51 e | 25.65 ± 4.11 g | 9.69 ± 2.50 g | 28.01 ± 5.04 g | −13.18 ± 2.15 d |
| Pantone 188 C | 26.69 ± 0.93 b,c | 14.08 ± 3.00 d | 3.81 ± 1.23 d,e | 14.59 ± 3.22 d,e | −18.17 ± 1.52 c |
| Pantone 187 C | 29.07 ± 1.97 d | 20.41 ± 3.99 f | 7.15 ± 2.36 f | 21.65 ± 4.53 f | −14.14 ± 2.55 d |
| Pantone 1817 C | 27.06 ± 1.06 c | 15.39 ± 3.90 e | 4.32 ± 1.54 e | 15.99 ± 4.17 e | −17.69 ± 1.56 c |
| Pantone 1807 C | **31.89** ± 3.02 f | **26.45** ± 4.05 g | **11.09** ± 3.46 h | **28.18** ± 4.64 g | **−10.17** ± 3.76 e |

### 3.2. Polyphenolic and Colour Analysis of Sour Cherry Fruits

TPC was determined with the CIELab values for each sour cherry accession (Table 5). Interestingly, the results showed that the accession with the lowest content of TPC was Kantorjanosi 3 (122.76 mgGAE/100 g fresh cherries), while 'Pipacs1' had the highest TPC (650.57 mgGAE/100 g fresh cherries). Furthermore, 'Pipacs 1' was the accession with the highest a* (26.45) and Chroma (28.18) values. 'Hortenzia Királynője' had the highest value of L*, b* and hue. Finally, 'Bosnyák' was the accession with the lowest L*, a*, b*, Chroma and hue of all the sour cherries analysed in this study.

**Table 5.** Total polyphenolic content (TPC) compared with CIELab values. Letters of the same column indicate significantly different values at $p < 0.05$. Bold and italic values represent highest and lowest, respectively. Asterisks (*) indicate the commercial cultivars.

| Accession | TPC | L* | a* | b* | Chroma | h$^{\pm33°}$ |
|---|---|---|---|---|---|---|
| Bosnyák | 283.59 ± 0.74 | *24.52* ± 0.78 a | 2.75 ± 1.15 a | *−0.05* ± 0.33 a | 2.77 ± 1.15 a | *−36.08* ± 6.97 a |
| Késői parasztmeggy | 220.11 ± 0.42 | 25.05 ± 1.20 a,b | 17.29 ± 3.46 m | 5.08 ± 1.51 k | 18.03 ± 3.75 k | −16.87 ± 1.35 h,i |
| Érdi Bőtermő* | 172.53 ± 0.69 | 25.36 ± 0.43 b,c | 7.35 ± 1.83 b | 1.50 ± 0.56 c | 7.50 ± 1.90 b | −21.74 ± 1.66 c |
| Érdi Jubileum* | 280.82 ± 0.77 | 25.66 ± 0.90 b,c,d | 4.21 ± 1.97 a | 0.68 ± 0.68 b | 4.28 ± 2.06 a | −25.08 ± 4.36 b |
| Korai Pándy | 289.19 ± 0.64 | 25.71 ± 0.87 c,d | 14.30 ± 1.51 h,i,j,k | 3.88 ± 0.60 f,g,h,i,j | 14.82 ± 1.62 g,h,i | −17.90 ± 0.86 e,f,g,h |
| Nagy Gobet | 329.83 ± 0.76 | 26.05 ± 0.64 d,e | 12.85 ± 2.60 d,e,f,g,h,i | 3.32 ± 0.99 d,e,f,g,h | 13.27 ± 2.76 d,e,f,g,h | −18.80 ± 1.48 e,f,g |
| Mogyoródi kései | 247.26 ± 0.95 | 26.24 ± 0.83 d,e,f | 12.53 ± 2.66 d,e,f,g,h | 3.23 ± 1.00 d,e,f,g,h | 12.94 ± 2.82 d,e,f,g | −18.86 ± 1.74 e,f,g |
| Korai cigány | 400.43 ± 1.24 | 26.27 ± 0.56 d,e,f | 9.19 ± 1.44 c | 2.07 ± 0.51 c | 9.43 ± 1.51 c | −20.43 ± 1.33 d |
| Pándy Bb. 119 * | 213.65 ± 0.98 | 26.40 ± 0.53 e,f,g | 11.46 ± 1.50 d,e | 2.93 ± 0.52 d,e | 11.83 ± 1.58 d | −18.75 ± 0.99 e,f,g |
| Dunabogdányi | 267.71 ± 0.29 | 26.40 ± 0.60 e,f,g | 11.79 ± 2.17 d,e,f | 2.97 ± 0.79 d,e | 12.16 ± 2.29 d,e | −19.03 ± 1.55 e,f,g |
| Pándy 43 * | 195.16 ± 1.11 | 26.43 ± 0.61 e,f,g | 13.56 ± 1.79 f,g,h,i,j,k | 3.52 ± 0.66 d,e,f,g,h,i | 14.01 ± 1.90 e,f,g,h,i | −18.57 ± 0.85 e,f,g |
| Pándy 279 * | 202.28 ± 0.94 | 26.43 ± 0.86 e,f,g | 12.14 ± 1.85 d,e,f,g | 3.14 ± 0.62 d,e,f,g | 12.54 ± 1.94 d,e,f | −18.55 ± 1.07 e,f,g |
| Velencei kései | 245.19 ± 1.01 | 26.50 ± 0.63 e,f,g | 13.18 ± 1.80 e,f,g,h,i,j | 3.48 ± 0.68 d,e,f,g,h,i | 13.63 ± 1.91 d,e,f,g,h,i | −18.30 ± 1.04 e,f,g,h |
| Cigány Késői | 270.39 ±2.87 | 26.51 ± 0.65 e,f,g | 12.72 ± 2.62 d,e,f,g,h,i | 3.34 ± 0.96 d,e,f,g,h,i | 13.15 ± 2.78 d,e,f,g | −18.51 ± 1.16 e,f,g |
| Késői cigány | 241.71 ± 0.47 | 26.51 ± 0.85 e,f,g | 12.38 ± 2.87 d,e,f,g | 3.13 ± 1.15 d,e,f,g | 12.78 ± 3.07 d,e,f | −19.15 ± 1.69 d,e,f |
| Cigánymeggy 7 * | 294.43 ± 0.49 | 26.52 ± 0.60 e,f,g | 11.68 ± 2.11 d,e | 2.99 ± 0.73 d,e,f | 12.06 ± 2.22 d,e | −18.81 ± 1.16 e,f,g |

**Table 5.** *Cont.*

| Accession | TPC | L* | a* | b* | Chroma | h$^{\pm 33°}$ |
|---|---|---|---|---|---|---|
| Fehérvári | 296.12 ± 0.70 | 26.55 ± 0.54 e,f,g | 13.56 ± 1.94 f,g,h,i,j,k | 3.61 ± 0.77 d,e,f,g,h,i | 14.03 ± 2.07 e,f,g,h,i | −18.23 ± 1.09 e,f,g,h |
| Édes Pipacs | 297.35 ± 0.73 | 26.69 ± 0.63 e,f,g | 13.81 ± 1.80 g,h,i,j,k | 3.74 ± 0.76 e,f,g,h,i | 14.31 ± 1.93 f,g,h,i | −17.98 ± 1.09 e,f,g,h |
| Újfehértói fürtös* | 466.19 ± 0.37 | 26.77 ± 0.61 e,f,g | 12.74 ± 1.86 d,e,f,g,h,i | 3.19 ± 0.67 d,e,f,g,h | 13.13 ± 1.96 d,e,f,g | −19.08 ± 1.09 d,e,f |
| Májusi hólyag | 256.16 ± 0.51 | 26.83 ± 0.72 f,g | 14.68 ± 2.08 j,k | 4.06 ± 0.87 h,i,j | 15.23 ± 2.24 h,i | −17.70 ± 1.22 f,g,h |
| Későn virágzó | 293.32 ± 0.74 | 26.92 ± 0.75 f,g | 15.01 ± 2.49 k,l | 4.22 ± 1.11 i,j | 15.60 ± 2.70 i,j | −17.51 ± 1.39 g,h |
| Bagi Meggy | 313.28 ± 0.75 | 26.94 ± 0.54 f,g | 11.30 ± 2.46 d | 2.81 ± 0.94 d | 11.65 ± 2.61 d | −19.43 ± 2.06 d,e |
| Péceli nagy | 281.76 ± 0.71 | 27.04 ± 0.54 g | 14.64 ± 1.53 j,k | 3.94 ± 0.56 g,h,i,j | 15.16 ± 1.62 h,i | −17.97 ± 0.69 e,f,g,h |
| Kántorjánosi 3 * | *122.76 ± 0.97* | 27.11 ± 0.99 g,h | 14.38 ± 3.45 i,j,k | 3.89 ± 1.40 f,g,h,i,j | 14.90 ± 3.70 g,h,i | −18.20 ± 1.43 e,f,g,h |
| Helyi Sötét | 315.29 ± 0.82 | 27.68 ± 1.37 h,i | 16.39 ± 5.11 l,m | 4.62 ± 2.24 j,k | 17.05 ± 5.52 j,k | −18.03 ± 2.93 e,f,g,h |
| Tiszabög 50/7 | 436.87 ± 0.86 | 27.94 ± 0.96 i | 18.91 ± 2.87 n | 5.84 ± 1.33 l | 19.80 ± 3.12 l | −16.04 ± 1.52 i |
| Fűzlevelű Kisszemű | 398.72 ± 0.42 | 28.25 ± 0.68 i | 19.89 ± 2.17 n | 6.12 ± 0.97 l | 20.81 ± 2.36 l | −16.00 ± 0.95 i |
| Szamosi meggy | 192.59 ± 0.93 | 30.21 ± 2.07 j | 21.91 ± 4.39 o | 8.46 ± 2.45 m | 23.50 ± 4.97 m | −12.24 ± 1.85 j |
| Pipacs1 * | **650.57 ± 1.41** | 30.72 ± 1.51 j | **26.45 ± 4.05 p** | 9.69 ± 2.50 n | **28.18 ± 4.64 n** | −13.18 ± 2.15 j |
| Hortenzia Királynője | 179.11 ± 0.93 | **31.89 ± 3.02 k** | 25.65 ± 4.11 p | **11.09 ± 3.46 o** | 28.01 ± 5.04 n | **−10.17 ± 3.76 k** |

### 3.3. Firmness, Soluble Solid Content (SSC) and Weight

Single accessions were described for their firmness, their soluble solid content, and fruit weight (Table 6). The firmest accession was 'Pipacs1′ (57.26 shore), whilst the least firm was 'Cigánymeggy 7′ (14.33 shore). Soluble solid content was the highest in 'Erdi Jubileum' (30.63% Brix), while the lowest was 'Kantorjanosi 3′ (19.17% Brix). The highest fruit weight was found in 'Mogyoródi kései' (6.62 g), while the lowest was in 'Helyi Sötét' (2.22 g). Seed weight was the smallest in 'Bagi meggy' (0.15 g) and the biggest in 'Pipacs1′ (0.52 g). Pulp weight was the highest in 'Mogyoródi Kései' (6.21 g) and the lowest in 'Helyi Sötét' (2 g).

**Table 6.** Total polyphenolic content (TPC) compared with fruit firmness, soluble solid content (SSC), and weight. Letters of the same column indicate significantly different values at $p < 0.05$. Bold and italic values represent highest and lowest, respectively. Asterisk (*) indicate the commercial cultivar.

| Accession | Firmness | Soluble Solid Content (SSC) | Fruit Weight | Seed Weight | Pulp Weight |
|---|---|---|---|---|---|
| Bagi Meggy | 23.38 ± 5.26 b,c,d,e | 26.37 ± 3.20 j | 2.38 ± 0.26 a | *0.15 ± 0.05 a* | 2.23 ± 0.26 a |
| Bosnyák | 32.45 ± 9.75 f,g,h,i,j | 25.90 ± 2.04 j | 3.49 ± 0.40 b,c | 0.19 ± 0.06 a,b | 3.30 ± 0.42 d,e,f |
| Cigány Késői | 22.16 ± 6.40 b,c,d | 19.60 ± 1.54 a,b,c | 3.38 ± 0.33 b,c | 0.28 ± 0.04 c,d,e | 3.11 ± 0.32 d,e,f |
| Cigánymeggy 7 * | *14.33 ± 5.77 a* | 23.24 ± 1.68 g,h,i | 3.50 ± 0.53 b,c | 0.30 ± 0.08 c,d,e,f | 3.21 ± 0.48 d,e,f |
| Dunabogdányi | 34.48 ± 5.68 i,j | 20.76 ± 1.44 a,b,c,d,e,f | 4.90 ± 0.31 e | 0.36 ± 0.08 e,f,g,h,i | 4.55 ± 0.30 h |
| Édes Pipacs | 37.42 ± 4.86 j | 21.94 ± 1.45 c,d,e,f,g,h | 4.91 ± 0.42 e | 0.37 ± 0.06 f,g,h,i,j | 4.54 ± 0.41 h |
| Érdi Bőtermő * | 25.99 ± 6.65 c,d,e,f | 19.97 ± 2.41 a,b,c,d | 5.80 ± 0.52 h,i | 0.32 ± 0.08 d,e,f,g,h | 5.48 ± 0.52 k,l |
| Érdi Jubileum * | 27.26 ± 6.29 d,e,f,g,h | **30.63 ± 2.98 l** | 3.71 ± 0.96 b,c,d | 0.28 ± 0.06 c,d,e | 3.43 ± 0.91 d,e,f,g |
| Fehérvári | 37.50 ± 4.86 j | 21.57 ± 1.17 b,c,d,e,f,g,h | 4.86 ± 0.42 e | 0.32 ± 0.07 d,e,f,g,h | 4.54 ± 0.41 h |
| Fűzlevelű Kisszemű | 44.54 ± 7.99 k,l | 23.05 ± 1.60 f,g,h | 2.71 ± 0.39 a | 0.24 ± 0.08 b,c,d | 2.47 ± 0.39 a,b,c |
| Helyi Sötét | 32.72 ± 9.07 f,g,h,i,j | 28.54 ± 2.50 k | 2.22 ± 0.19 a | 0.23 ± 0.06 b,c | *2.00 ± 0.18 a* |
| Hortenzia Királynője | 19.99 ± 6.07 a,b,c | 25.21 ± 2.55 i,j | 2.60 ± 0.31 a | 0.22 ± 0.06 b,c | 2.38 ± 0.32 a,b |
| Kántorjánosi 3 * | 33.39 ± 4.52 g,h,i,j | *19.17 ± 2.20 a* | 5.42 ± 0.75 e,f,g,h | 0.45 ± 0.08 j,k | 4.97 ± 0.71 h,i,j |
| Késői cigány | 18.75 ± 4.92 a,b | 22.71 ± 2.49 e,f,g,h | 3.23 ± 0.34 b | 0.38 ± 0.06 g,h,i,j,k | 2.85 ± 0.32 b,c,d |
| Késői parasztmeggy | 37.00 ± 7.20 i,j | 20.93 ± 1.97 a,b,c,d,e,f | 5.28 ± 0.63 e,f,g,h | 0.37 ± 0.07 f,g,h,i | 4.92 ± 0.63 h,i |
| Későn virágzó | 26.32 ± 6.94 c,d,e,f,g | 23.70 ± 1.80 h,i | 3.82 ± 0.55 b,c,d | 0.25 ± 0.08 b,c,d | 3.57 ± 0.51 f,g |
| Korai cigány | 38.39 ± 6.63 j,k | 22.88 ± 1.97 f,g,h | 3.27 ± 0.27 b,c | 0.32 ± 0.07 d,e,f,g,h | 2.95 ± 0.28 c,d,e |
| Korai Pándy | 38.29 ± 4.84 j,k | 20.87 ± 1.85 a,b,c,d,e,f | 3.73 ± 0.39 b,c,d | 0.28 ± 0.04 c,d,e | 3.45 ± 0.39 e,f,g, |

**Table 6.** *Cont.*

| Accession | Firmness | Soluble Solid Content (SSC) | Fruit Weight | Seed Weight | Pulp Weight |
|---|---|---|---|---|---|
| Májusi hólyag | 33.88 ± 6.30 h,i,j | 20.36 ± 1.41 a,b,c,d | 5.01 ± 0.62 e,f | 0.35 ± 0.07 e,f,g,h,i | 4.67 ± 0.59 h |
| Mogyoródi kései | 37.39 ± 4.90 j | 22.27 ± 1.67 d,e,f,g,h | **6.62** ± 0.74 j | 0.41 ± 0.09 i,j,k | **6.21** ± 0.71 m |
| Nagy Gobet | 37.34 ± 4.53 j | 20.42 ± 1.38 a,b,c,d,e | 6.29 ± 0.86 i,j | 0.33 ± 0.06 e,f,g,h,i | 5.96 ± 0.84 l,m |
| Pándy 279 * | 32.30 ± 5.30 f,g,h,i,j | 19.51 ± 2.05 a,b | 5.72 ± 0.69 g,h | 0.45 ± 0.09 k,l | 5.27 ± 0.70 i,j |
| Pándy 43 * | 36.12 ± 4.86 i,j | 20.87 ± 1.28 a,b,c,d,e,f | 4.82 ± 0.66 e | 0.28 ± 0.08 c,d,e | 4.54 ± 0.65 h |
| Pándy Bb. 119 * | 32.15 ± 4.53 f,g,h,i,j | 19.61 ± 1.13 a,b,c | 5.23 ± 0.49 e,f,g,h | 0.39 ± 0.09 g,h,i,j,k | 4.84 ± 0.45 h,i |
| Péceli nagy | 37.47 ± 6.60 j | 21.02 ± 3.01 a,b,c,d,e,f,g | 4.24 ± 0.42 d | 0.31 ± 0.04 d,e,f,g | 3.93 ± 0.42 g |
| Pipacs1 * | **57.26** ± 11.73 m | 26.01 ± 2.16 j | 3.87 ± 0.56 c,d | **0.52** ± 0.12 l | 3.35 ± 0.48 d,e,f |
| Szamosi meggy | 29.87 ± 5.51 e,f,g,h,i | 26.44 ± 3.69 j | 5.57 ± 0.85 f,g,h | 0.33 ± 0.06 e,f,g,h,i | 5.24 ± 0.83 i,j |
| Tiszabög 50/7 | 46.96 ± 7.19 l | 21.22 ± 1.69 a,b,c,d,e,f,g | 6.39 ± 0.65 j | 0.40 ± 0.07 h,i,j,k | 5.99 ± 0.65 l,m |
| Újfehértói fürtös * | 36.37 ± 8.11 i,j | 22.75 ± 1.68 e,f,g,h | 3.55 ± 0.55 b,c | 0.32 ± 0.04 d,e,f,g,h | 3.23 ± 0.53 d,e,f |
| Velencei kései | 35.33 ± 4.99 i,j | 20.53 ± 1.04 a,b,c,d,e | 5.19 ± 0.77 e,f,g | 0.29 ± 0.06 c,d,e | 4.91 ± 0.77 h,i |

*3.4. Principal Component Analysis (PCA) and Bivariate Analysis*

Bivariate analysis was performed between each component. It is possible to observe colorimetric data such as L*, a*, b*, Chroma, and hue positively correlating with one another, as expected (Table 7). Negative correlation is observed between CIELab, Pantone, and Ctifl. Interestingly, TPC positively correlates with SSC, firmness, and acidity. On the other hand, TPC negatively correlates with sweetness and weight-related characteristics, particularly with fruit weight. Bivariate analysis correlated TPC positively with L*, a*, b*, and Chroma but, interestingly, not with hue. SSC negatively correlates with acidity, juiciness, and weight, while it correlates positively with colorimetric and TPC data.

**Table 7.** Bivariate correlation with Spearman's rho. *p* value is considered significant at <0.05.

| | | L | a | b | C | $h^{\pm 33°}$ | TPC | UPOV Skin Colour | Ctifl Scale | UPOV Flesh Colour | UPOV Juice Colour | Soluble Solid Content (SSC) | Firmness (d) Turoni | Acidity | Sweetness | Juiciness | Weight (g) | Seed Weight (g) | Fruit Pulp (g) |
|---|---|---|---|---|---|---|---|---|---|---|---|---|---|---|---|---|---|---|---|
| Spearman's rho | L | 1.000 | | | | | | | | | | | | | | | | | |
| | a | 0.694 ** | 1.000 | | | | | | | | | | | | | | | | |
| | b | 0.709 ** | 0.992 ** | 1.000 | | | | | | | | | | | | | | | |
| | C | 0.696 ** | 1.000 ** | 0.993 ** | 1.000 | | | | | | | | | | | | | | |
| | $h^{\pm 33°}$ | 0.718 ** | 0.922 ** | 0.960 ** | 0.926 ** | 1.000 | | | | | | | | | | | | | |
| | TPC | 0.281 ** | 0.296 ** | 0.266 * | 0.291 ** | 0.196 | 1.000 | | | | | | | | | | | | |
| | UPOV Skin colour | −0.322 ** | −0.493 ** | −0.502 ** | −0.494 ** | −0.495 ** | 0.191 | 1.000 | | | | | | | | | | | |
| | Ctifl scale | −0.314 ** | −0.518 ** | −0.524 ** | −0.519 ** | −0.507 ** | 0.117 | 0.677 ** | 1.000 | | | | | | | | | | |
| | UPOV Flesh colour | −0.325 ** | −0.533 ** | −0.547 ** | −0.535 ** | −0.553 ** | 0.143 | 0.802 ** | 0.700 ** | 1.000 | | | | | | | | | |
| | UPOV Juice colour | −0.322 ** | −0.542 ** | −0.552 ** | −0.544 ** | −0.546 ** | 0.139 | 0.753 ** | 0.666 ** | 0.885 ** | 1.000 | | | | | | | | |
| | Soluble solid content (SSC) | 0.236 ** | 0.125 ** | 0.120 ** | 0.124 ** | 0.108 ** | 0.294 ** | 0.116 ** | 0.213 ** | 0.098 * | 0.033 | 1.000 | | | | | | | |
| | Firmness (d) Turoni | 0.054 | 0.244 ** | 0.232 ** | 0.243 ** | 0.196 ** | 0.444 ** | −0.255 ** | −0.275 ** | −0.312 ** | −0.242 ** | −0.023 | 1.000 | | | | | | |
| | Acidity | 0.161 ** | 0.208 ** | 0.205 ** | 0.207 ** | 0.195 ** | 0.333 ** | −0.219 ** | −0.298 ** | −0.154 ** | −0.170 ** | −0.121 ** | 0.347 ** | 1.000 | | | | | |
| | Sweetness | −0.012 | 0.000 | 0.026 | 0.004 | 0.066 | −0.403 ** | −0.115 ** | −0.098 * | 0.005 | 0.028 | 0.062 | −0.155 ** | −0.032 | 1.000 | | | | |
| | Juiciness | 0.309 ** | 0.397 ** | 0.405 ** | 0.398 ** | 0.407 ** | −0.107 | −0.605 ** | −0.559 ** | −0.625 ** | −0.445 ** | −0.189 ** | 0.217 ** | 0.280 ** | 0.196 ** | 1.000 | | | |
| | Weight (g) | −0.219 ** | −0.068 | −0.050 | −0.066 | −0.016 | −0.409 ** | −0.404 ** | −0.277 ** | −0.312 ** | −0.281 ** | −0.473 ** | 0.199 ** | 0.297 ** | 0.186 ** | 0.267 ** | 1.000 | | |
| | seed weight (g) | −0.050 | 0.018 | 0.023 | 0.019 | 0.040 | −0.232 * | −0.343 ** | −0.277 ** | −0.250 ** | −0.220 ** | −0.270 ** | 0.219 ** | 0.396 ** | −0.032 | 0.227 ** | 0.523 ** | 1.000 | |
| | Fruit pulp (g) | −0.229 ** | −0.078 | −0.059 | −0.076 | −0.025 | −0.409 ** | −0.386 ** | −0.262 ** | −0.300 ** | −0.268 ** | −0.473 ** | 0.188 ** | 0.280 ** | 0.199 ** | 0.261 ** | 0.998 ** | 0.467 ** | 1.000 |

** Correlation is significant at the 0.01 level (2-tailed). * Correlation is significant at the 0.05 level (2-tailed).

PCA indicated a positive correlation between L*, a*, b*, Chroma, hue, and juiciness in the first component (Table 8). The second component analysis positively correlated TPC with L*, a*, b*, Chroma, and SSC. The third component analysis positively correlated firmness, TPC, and acidity. The fourth component analysis positively correlated sweetness with SSC. From the obtained PC analyses, we created a 2D scatterplot where PC1, PC2, PC3, and PC4 are considered and we plot single components with different colours. Similarly, we plot the different accessions based on the previously performed PC analysis (Figures 1 and 2).

**Table 8.** Principal component analysis of colour, physical characteristics, flavour profile, and fruit weight.

| | Component Matrix [a] | | | |
| --- | --- | --- | --- | --- |
| | **Component** | | | |
| | **PC1** | **PC2** | **PC3** | **PC4** |
| a | 0.922 | 0.298 | 0.01 | 0.029 |
| C | 0.922 | 0.305 | −0.007 | 0.036 |
| h33+− | 0.914 | 0.092 | −0.17 | −0.023 |
| b | 0.905 | 0.333 | −0.11 | 0.078 |
| UPOV Flesh colour | −0.854 | 0.283 | 0.104 | 0.09 |
| UPOV Skin colour | −0.853 | 0.305 | −0.006 | 0.012 |
| UPOV Juice colour | −0.85 | 0.225 | 0.113 | 0.057 |
| Ctifl scale | −0.808 | 0.277 | 0.061 | 0.077 |
| L | 0.791 | 0.432 | −0.162 | 0.072 |
| Juiciness | 0.644 | −0.347 | −0.148 | 0.056 |
| Weight (g) | 0.1 | −0.911 | 0.065 | 0.046 |
| Fruit pulp (g) | 0.095 | −0.898 | 0.048 | 0.029 |
| seed weight (g) | 0.111 | −0.545 | 0.267 | 0.263 |
| Firmness (d) Turoni | 0.256 | −0.042 | 0.753 | −0.014 |
| TPC | 0.284 | 0.439 | 0.717 | 0.05 |
| Acidity | 0.292 | −0.292 | 0.606 | 0.305 |
| Sweetness | −0.21 | −0.316 | −0.431 | 0.664 |
| Soluble solid content (SSC) | −0.001 | 0.598 | 0.01 | 0.617 |

[a] Extraction Method: Principal Component Analysis. 4 components extracted.

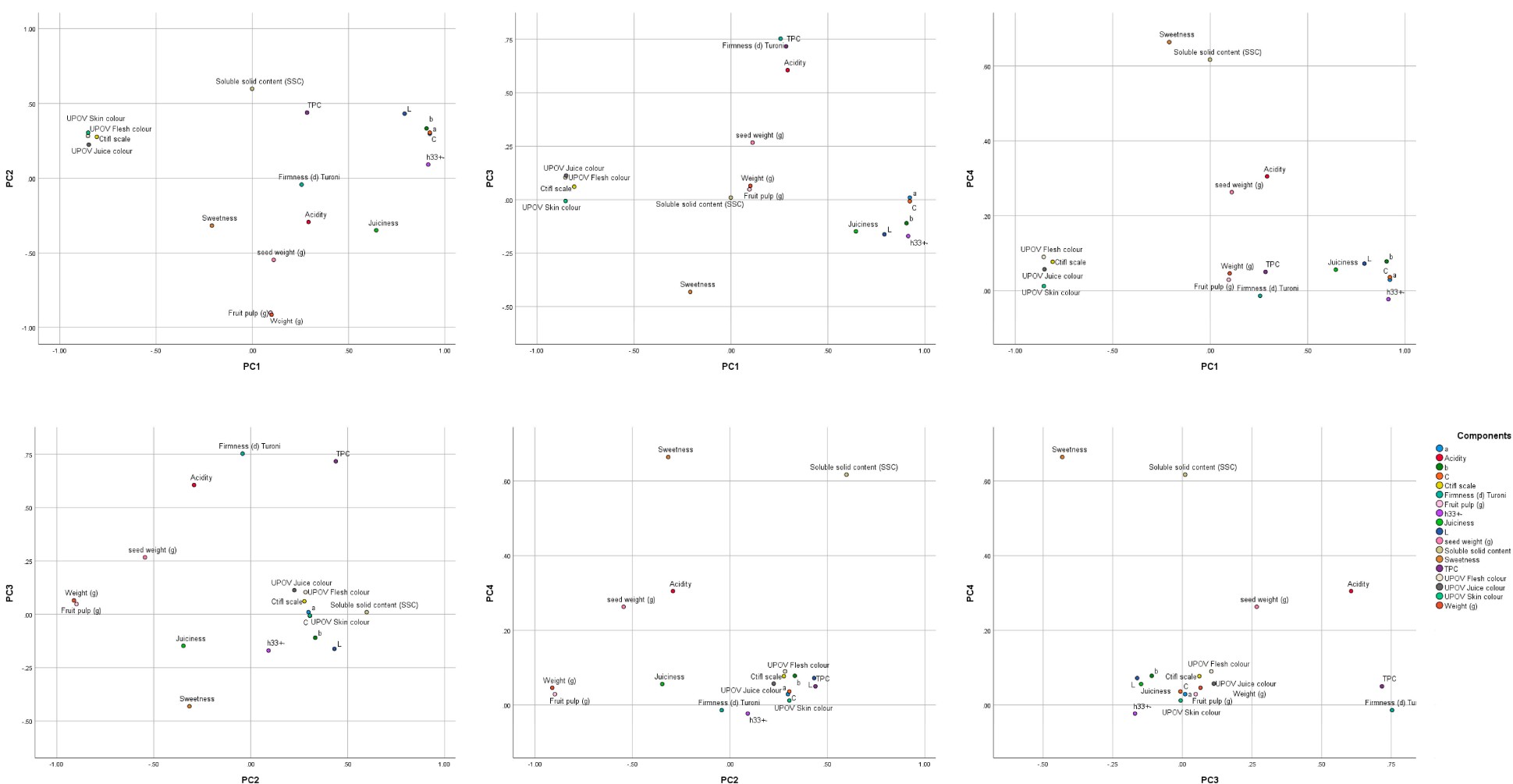

**Figure 1.** 2d scatterplot of the principal component analysis. Compared characteristics are indicated with different colours.

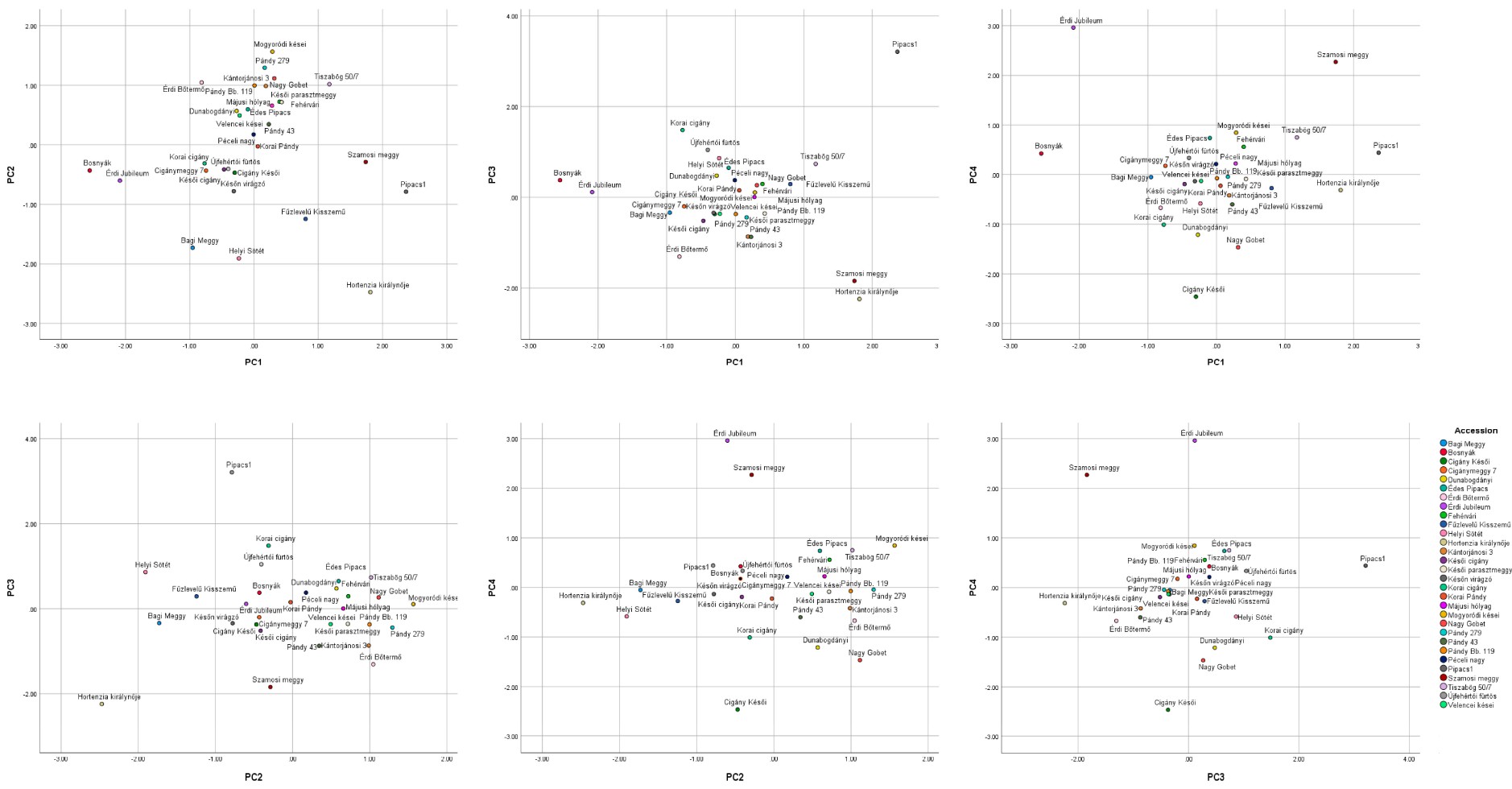

**Figure 2.** 2d Scatterplot of principal component analysis performed according to different accession tested. Different colours indicate different varieties.

## 4. Discussion

Total polyphenolic content was compared to CIELab values to evaluate the correlation between the polyphenols and colour of sour cherries. According to previous literature, 'Érdi botermő', 'Üjfehértói furtös', and 'Pipacs1′ had similar results regarding the TPC values indicated in this study [8]. In this analysis, the highest TPC was found in 'Pipacs1′ (650.5 mgGAE/100 g fresh cherries), while the lowest was found in 'Kantorjanosi 3′ (122.7 mgGAE/100 g fresh cherries). 'Érdi botermő' and 'Érdi Jubileum' had low TPC content as well (172.53 and 280.82 mgGAE/100 g fresh cherries, respectively), in line with previous research [8,15–18]. Fruit colour measured with CIELab values suggested that the darkest accession was 'Bosnyák' overall, whilst the lightest was 'Hortenzia Királynője'. 'Bosnyák' was referred to as a very dark fruit in previous research, when a population was compared for its antioxidant and anthocyanin content [19]. In their study, Veres and colleagues found that dark varieties such as 'Bosnyák' had high melatonin accumulation, making it a good candidate for further studies. Colorimetric data coincide with several previously published papers. Viljevac and colleagues [18] in their research indicated similar values for the 'Cigany' cultivar as 'Cigánymeggy 7′ in our research, thus, possibly referring to the same accession. 'Érdi Jubileum' was indicated as a very firm accession in a previous study [15]. This characteristic is very important for the marketability of sour cherries in the local and even more in international markets, making the fruit suitable for long distance travel. 'Érdi Jubileum' and 'Érdi botermö' were both tested in a previous study, where weight and firmness of the fruit were comparable to results of our study. In our study however, 'Pipacs1′ was indicated as the firmest accession. Principal component analysis between categories indicated that there is a positive correlation between L*, a*, b*, Chroma, and hue, as expected. Negative correlation is shown between Ctifl, Pantone, and UPOV fruit skin colour, as expected, since all three scaling systems have a higher score for darker colours, while CIELab is at the opposite end. Pantone appears to be a useful alternative to categorize fruit skin colour. A palette of colours could be derived from the indicated values and created specifically for Hungarian sour cherry collections to help breeders evaluate using a scaling system from light pink (1) to black (10). In contrast with Viljevac [18], no negative correlation was found comparing TPC with L*, a*, b*, Chroma, or hue. Instead, as indicated by Najafzadeh and colleagues [15], a positive correlation was observed between TPC and fruit firmness. Furthermore, this study found the same negative correlation between total polyphenolic content and total sugar as observed by Najafzadeh and colleagues, while in our study, a negative correlation between TPC and SSC appear similar. A negative correlation was also observed between TPC and fruit weight. The highest fruit weight was recorded for 'Mogyoródi kései' (6.62 g). This variety could be investigated further for selection of fruit size in the future and integration in the breeding program. The TPC content in fruit was comparatively similarly to other studies, as previously indicated. However, fruit development may be influenced by other factors such as yearly rainfall, soil nutrient availability, and temperature. Genetic background information may influence fruit development as well, indicating that fruits from different accessions may differ from other grown in different countries [20,21].

## 5. Conclusions

TPC has an important role in fruit development, and in the future, it would be important to evaluate other components such as antioxidant activity and volatile compounds. Colorimetric and TPC analysis suggested possible future candidates for breeding programmes. 'Mogyoródi kései', for example, appears to be not only very big but also an accession with interestingly low TPC values. Furthermore, the colour is quite dark, which is appreciated by local Hungarian consumers. 'Bosnyák', with its high content of SSC as well as 'Pipacs1′ with its high TPC values and interesting flavour profile would be good candidates for production as conserved food items. Followup studies on anthocyanin, sugar content, and the possible identification of commercial purposes will follow, to select and integrate the most successful varieties in the next breeding programme. Data

collected indicated not only that the TPC and colour positively correlated, but also that TPC positively correlates with acidity and firmness. Further studies will be conducted to evaluate the correlation between soil, climatic factors, and genetic profile over the chemical composition of Hungarian sour cherry.

**Supplementary Materials:** The following supporting information can be downloaded at: https://www.mdpi.com/article/10.3390/agriculture13071287/s1. Fruit laboratory and field pictures of the sour cherry analyzed in this study. Pictures are arranged according to the colour shades, from darkest to lightest, and are shown from top left to bottom right.

**Author Contributions:** Conceptualization F.D.: Conceptualization, Methodology, Software, Validation, Formal Analysis, Investigation, Writing—Original Draft Preparation, Visualization Funding Acquisition, Project Administration; S.S.: Writing—Review Editing, Visualization; Z.B.: Writing—Review and Editing, Visualization, Supervision, Project Administration; G.B.: Writing—Review and Editing, Visualization; V.U.: Writing—Review and Editing; B.M.: Writing—Review and Editing; C.M.: TPC Formal Analysis Investigation; L.G.: Methodology, Software, Validation, Formal Analysis Investigation, Visualization, Supervision, Project Administration. All authors have read and agreed to the published version of the manuscript.

**Funding:** This research was funded by: Danube Rectors' Conference. Grant Number: DRC Initiative Fund.

**Institutional Review Board Statement:** Not applicable.

**Data Availability Statement:** Not applicable.

**Conflicts of Interest:** The authors declare no conflict of interest.

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
