# Peer review of "Polyphenolic and Fruit Colorimetric Analysis of Hungarian Sour Cherry Genebank Accessions"

_agriculture, doi:10.3390/agriculture13071287_

Round 1

Reviewer 1 Report

Desiderio et al. have investigated the fruit polyphenols and colour of Hungarian sour cherry gene bank accessions. As per my opinion, although study is preliminary research in this field, have significance with respect to Agriculture journal. Some of the flaws are enlisted below;

1.      In abstract, rather than focusing on methodology a lot, authors should explain more results.

2.      The results should be discussed with numerical evidences.

3.      Keyword’s usefulness is to make easier the search of the article using the most common scientific search engines. Since several keywords are or repeated several times in the abstract, I strongly advise the authors to change some of the proposed keywords with new ones.

4.      I advise authors to deeply study these variables on molecular level. They should know the mechanisms behind polyphenol profiles.

5.      The citation style should be rechecked. Line 195.

6.      The discussion part is also needed to be improved.

English language is fine

Reviewer 2 Report

The objective of this article was to compare varieties according to different quality parameters and to find significant correlations between the analyzed parameters.

The statistical analysis and results should be according to the objective of the study. In my opinion, the description of statistical analysis is not clear according to the presentation of results. For exemple, the autorhs indicate that L*a*b*C*h were compared each to toal polyphenolic content and then these last ones were compared to fruit firmness. However, the results only show differences between cultivars in these parameters and no correlations between theses parameters are shown.

From the objective described in the manuscript, I recommend to using other statistical methods (lineal correlations, multivariant analysis, PCA, PLS,...) in the data analysis and rewritting the manuscript according to the results.

Reviewer 3 Report

The subject of the manuscript is quite interesting in the selection of promising sour cherry materials with high levels of polyphenols and their importance for consumption and in the European market. However, I suggest some corrections and suggestions for improvement in the work and they must be accepted in order to be ready for publication.

Round 2

Reviewer 1 Report

Authors just added the multivariate analysis as requested by another reviewer. However, I requested them to focus on molecular studies. Furthermore, I can't see any changes in abstract section. 

Language is fine
